# Quality and Quantity of Protein Intake Influence Incidence of Type 2 Diabetes Mellitus in Coronary Heart Disease Patients: From the CORDIOPREV Study

**DOI:** 10.3390/nu13041217

**Published:** 2021-04-07

**Authors:** Silvia de la Cruz-Ares, Francisco M. Gutiérrez-Mariscal, Juan F. Alcalá-Díaz, Gracia M. Quintana-Navarro, Alicia Podadera-Herreros, Magdalena P. Cardelo, José D. Torres-Peña, Antonio P. Arenas-de Larriva, Pablo Pérez-Martínez, Javier Delgado-Lista, Elena M. Yubero-Serrano, José López-Miranda

**Affiliations:** 1Lipids and Atherosclerosis Unit, Unidad de Gestión Clínica de Medicina Interna, Maimonides Institute for Biomedical Research in Córdoba, Reina Sofia University Hospital, University of Córdoba, 14004 Córdoba, Spain; scruz@uco.es (S.d.l.C.-A.); fmgutierrezm@hotmail.com (F.M.G.-M.); jfalcala@gmail.com (J.F.A.-D.); g.quintana.navarro@gmail.com (G.M.Q.-N.); alicia.podadera@gmail.com (A.P.-H.); malenipc023@gmail.com (M.P.C.); azarel_00@hotmail.com (J.D.T.-P.); aparenaslarriva@gmail.com (A.P.A.-d.L.); pablopermar@yahoo.es (P.P.-M.); delgadolista@gmail.com (J.D.-L.); 2CIBER Fisiopatología Obesidad y Nutrición (CIBEROBN), Instituto de Salud Carlos III, 14004 Córdoba, Spain

**Keywords:** plant proteins, type 2 diabetes mellitus, cardiovascular disease

## Abstract

Evidence suggests that enriching a diet with plant-based proteins could reduce the risk of developing type 2 diabetes mellitus. In the present work, we evaluated the association between the change in plant protein intake (adjusted by energy) and incidence of type 2 diabetes mellitus in patients with coronary heart disease from the CORDIOPREV (coronary diet intervention with olive oil and cardiovascular prevention) study. At baseline and during the follow-up, patients underwent medical examination and blood and oral glucose tolerance tests. Information on patient’s dietary intake was gathered by registered dietitians using a validated food frequency questionnaire. A total of 106 out of 436 nondiabetic patients at baseline developed type 2 diabetes mellitus after a median follow-up of 60 months. Cox regression analyses showed that patients who belonged to the group that increased plant protein intake exhibited a lower risk of developing the disease (HR = 0.64, (0.43–0.96)). Changes in plant protein intake were positively correlated with changes in carbohydrates, fibre, and legumes intake and negatively correlated with changes in saturated fatty acids intake. Results of the present study support the need of improving diet with plant-based proteins to prevent the onset of type 2 diabetes mellitus.

## 1. Introduction

Cardiovascular disease (CVD) and type 2 diabetes mellitus (T2DM) constitute two of the leading causes of disability and mortality worldwide, bringing on important socioeconomic implications in terms of potential and productive life lost and healthcare costs [1]. Importantly, when both conditions are simultaneously present, the risk and severity of future cardiovascular events is considerably increased [2], and it can be further aggravated with the presence of metabolic risk factors such as hypertension, dyslipidaemia, or obesity [3]. Therefore, searching for new therapeutic strategies to prevent or delay the onset of T2DM, particularly in CVD patients, is of vital importance in cardiovascular research programs. In this sense, a large number of studies suggest that lifestyle changes (increased physical activity, adherence to a healthy diet, avoidance of smoking, and maintenance of an adequate body weight) might help to prevent the onset of both conditions [4]. Concerning dietary interventions, there is compelling evidence that food ingredients might have an impact on glucose regulation, as some food components improve β-cell function, insulin sensitivity, glycaemic control, or oxidative stress [5]. In this regard, frequent intake of red meat and processed meat has been associated with an increased risk of developing T2DM, whereas whole-grain and fibre intake has showed an inverse association with T2DM incidence [6]. When it comes to macronutrient composition, there is not consensus on the optimal proportions of carbohydrates, proteins, and fats to prevent T2DM or halt its progression. Although increasing protein intake has been proposed to induce weight loss and to improve insulin sensitivity, in the long-term, high-protein diets have been associated with an increased risk of T2DM [7]. Nonetheless, the observed connection is largely attributed to protein of animal origin, which also has been linked all-cause, cancer-related, and CVD-related mortality [7]. In fact, different studies have reported that replacing 5% of energy from animal protein with plant protein is associated with a reduction of T2DM risk [8], that adopting dietary patterns that favours protein from vegetal origin might help decline the incidence of T2DM [9], and that enriching diet with plant-based proteins is thought to promote long-term health and longevity [10].

Based on the previous evidence, the objective of this research work was to evaluate, within the framework of the CORDIOPREV (coronary diet intervention with olive oil and cardiovascular prevention) study, whether changes in dietary habits towards the consumption of more plant-based proteins, at the expense of proteins from animal sources, were associated with a reduction of T2DM incidence in a population with coronary heart disease.

## 2. Methods

### 2.1. Study Population

The present research work has been conducted within the framework of the CORDIOPREV (coronary diet intervention with olive oil and cardiovascular prevention) study (clinicaltrials.gov number NCT00924937, ethic approval number 1496/27/03/2009), a randomized, single-blind, controlled dietary intervention trial in 1002 coronary heart disease patients of European ancestry, who had their last coronary event more than 6 months before enrolment. Rationale, methodology, and baseline characteristics of study participants have been published elsewhere [11]. At baseline and during every year of follow-up study, participants passed medical and dietary interviews and underwent blood and oral glucose tolerance tests (OGTT). Trial protocol and all amendments were approved by the local ethics committees, following the Helsinki Declaration and Good Clinical Practice guidelines. All participants gave written informed consent to participate in the study.

For the present work, 540 CORDIOPREV participants who had a medical record of T2DM received glucose-lowering therapy and/or presented any of the American Diabetes Association (ADA) criteria for diagnosis of T2DM (fasting glucose ≥ 126 mg/dL, 2 h glucose during OGTT ≥ 200 mg/dL, or HbA1c ≥ 6.5%) [12] at baseline were excluded from the analysis. Of the remaining 462 participants free of T2DM at baseline, we included in this substudy those subjects who completed at least one year of follow-up (*n* = 437). From this group, a total of 107 subjects developed T2DM according to all the ADA diagnosis criteria, evaluated on the basis of OGTT performed each year during the median follow-up of 60 months. However, of the 107 patients who developed T2DM, one was excluded for not meeting a total energy intake inside the prespecified range (>500 kcal/day or <3500 kcal/day for women and >800 kcal/day or <4000 kcal/day for men [13]) in the food-frequency questionnaires (FFQ). Thus, for this substudy, the final number of patients who developed T2DM was 106 out of 436 non T2DM patients at baseline from the CORDIOPREV study.

### 2.2. Dietary Intervention

The primary goal of CORDIOPREV study was to modify eating behaviour of participants towards the allocated diet (low-fat or Med diet), without promoting calorie restriction, weight loss, or physical activity. Participants in both intervention groups received the same intensive dietary counselling. Both diets included foods from all major food groups: Med diet comprised a minimum of 35% energy from total fat (22% monounsaturated fatty acids-MUFAs, 6% polyunsaturated fatty acids-PUFAs, and <10% saturated fatty acids-SFAs), 15% from protein, and a maximum of 50% energy from carbohydrate. Low-fat diet consisted of <30% energy from total fat (12–14% MUFAs, 6–8% PUFAs and <10% SFAs), 15% from protein, and a minimum of 55% energy from carbohydrate.

### 2.3. Dietary Intake Assessment

Information on habitual dietary intake was gathered by registered dietitians at baseline and at each year of follow-up using a validated 137 item semiquantitative FFQ. To complete the FFQ, patients reported their average intake of different foods and beverages over the previous 12 months. Consumption frequencies were registered in nine categories ranging from “never or less than one time per month” to “six or more times per day”. Energy and nutrient intake were calculated using the Spanish Food Composition Tables [14,15]. Percentage of contribution of foods and nutrients to the mean daily energy intake was also calculated. A more detailed description of the dietary intervention is shown in Quintana-Navarro et al. (2019) [16]. To investigate the association between changes in the consumption of plant-based proteins at the expense of proteins from animal source and T2DM incidence, study participants were classified into two groups according to the median of change (Δ) in plant protein intake, expressed as percentage of energy (% E) (changes produced between post- and pre-intervention, calculated as the value of percentage of energy from plant protein after one year of dietary intervention minus the value at baseline), regardless of the type of consumed diet. Thus, patients whose Δ in plant-based protein consumption was below the median were sorted into the group which decreased plant protein intake (*n* = 218), and patients whose Δ in plant-based protein consumption was above the median were sorted into the group which increased plant protein intake (*n* = 218). With the former classification, it was possible to select study participants whose dietary habits were more noticeably modified towards a shift in plant protein intake after one year of follow-up thanks to dietary intervention.

### 2.4. Laboratory Analysis

Following a 12 h fasting period, patients were admitted to the laboratory for anthropometric and biochemical test (body mass index (BMI), waist circumference, systolic blood pressure, diastolic blood pressure, HDL-cholesterol (c-HDL), LDL-cholesterol (c-LDL), triglycerides (TG), total cholesterol, highly sensitive C-reactive protein (hs-CRP), glucose, and glycated haemoglobin (HbA1c)). Smoking status, alcohol intake, and drug therapy were also registered for each participant.

Anthropometric parameters were measured by trained dietitians using calibrated scales (BF511 body composition analyzer/scale, OMROM, Japan) and a wall-mounted stadiometer (Seca 242, HealthCheck Systems, Brooklyn, NY). Waist circumference was measured midway between the lowest rib and the iliac crest. BMI was calculated as weight per square meter (kg/m^2^). Blood pressure was measured with a validated digital automated blood pressure monitor, and hypertension was defined as a systolic blood pressure ≥130 mmHg, diastolic blood pressure ≥85 mmHg, and/or current use of antihypertensive agents.

Venous blood samples were collected from the antecubital vein in VacutainerTM tubes containing EDTA or no anticoagulant. Serum parameters were measured by spectrophotometry using an Architect c-16000 analyser (Abbott^®^, Chicago, IL, USA): hexokinase method for glucose and oxidation-peroxidation for c-HDL, total cholesterol, and TG. C-LDL was calculated using the Friedewald formula provided serum TG levels were <400 mg/dL. Plasma levels of insulin were measured by chemiluminescent microparticle immunoassay using an i-2000 Abbott Architect^®^ analyser. The plasma concentrations of hs-CRP were determined by high sensitivity ELISA (BioCheck, Inc., Foster City, CA, USA).

Patients underwent a standard OGTT at baseline and every year of follow-up. OGTT was performed following a standard procedure [17]. In brief, patients were asked to fast for 12 h (from foods and drugs), to refrain from smoking during the fasting period, to avoid strenuous physical activity the day before the test, and not to consume alcohol during the previous 7 days. At 8.00 a.m., patients were admitted to the laboratory, where a sample of blood was taken (0 min), and then, after a 75 g of flavoured glucose load (Trutol 75, Custom Laboratories, Baltimore, MD, USA), blood was sampled at 30, 60, 90, and 120 min to determine plasma glucose and insulin levels. Homeostasis model assessment of insulin resistance (HOMA-IR) was derived from fasting insulin (μU L^−1^) × fasting glucose (μmoles L^−1^)/22.5.

T2DM onset during follow-up was diagnosed by internal medicine physicians based on glucose tolerance tests performed each year. Diabetes status was considered when patients started undergoing glucose-lowering therapy and/or exhibited some of the ADA T2DM diagnostic criteria [12] during follow-up visits.

### 2.5. Statistical Analysis

Baseline characteristics are presented as mean (standard error of mean–SEM) for quantitative variables, percentages (%), and numbers (*n*) for categorical variables. Shapiro–Wilk normality test was performed to assess normality in continuous variables, and nonnormally distributed ones were sqrt-transformed. Unpaired t-test, Wilcoxon rank sum test for data that did not fit normal distribution after sqrt-transformation, and Chi-square test were employed to assess differences between groups. Paired *t*-test and Wilcoxon signed rank test for not normally distributed data were used to assess within group differences. Correlations between the Δ (1 year postintervention minus baseline values) in energy, nutrients, and food intake were calculated by means of a Spearman’s rank correlation. Probability of T2DM incidence was calculated using the Kaplan–Meier method of estimating the cumulative probability of an event in the group of those who had a Δ in plant protein (% E) intake above or below the median of the population after receiving dietary counselling (median = 0.248%). Time-dependent Cox regression models were used to identify significant factors associated with the time of incidence (full model was implemented with the following variables: sex, age, intervention group, prevalence of hypertension, baseline levels of c-HDL, TG and BMI, statin use, smoking status (never, former, or current smoker), and alcohol intake). Statistical analyses were performed using R version 4.0.3 software (R Foundation for Statistical Computing, Vienna, Austria) [18] on the RStudio platform and Tidyverse, Hmisc, corrplot, survival, and survminer R packages.

## 3. Results

### 3.1. Baseline Characteristics of Study Population

Table 1 shows baseline anthropometric and biochemical characteristics of patients classified according to the median of Δ in plant protein consumption (expressed as % E). Patients who increased plant protein intake exhibited slightly lower Apo A1, HbA1c, and HOMA-IR values than patients who decreased plant protein intake (all *p* < 0.05). They also exhibited a larger percentage of patients undergoing treatment with statins (*p* < 0.01). No statistically significant differences were observed in the remaining clinical parameters. The number of patients following each diet (low-fat or Med diet) were homogeneously distributed between the two groups of Δ in plant protein intake.

### 3.2. Plant Protein Intake after Dietary Intervention

Participants belonging to the group of patients who decreased plant protein intake did not significantly modify the amount of plant protein in their diet compared to baseline, but for the first year of follow-up, in which it decreased (Figure 1). On the other hand, participants belonging to the group of patients who increased plant protein intake augmented the percentage of energy obtained from plant proteins during follow-up with respect to baseline and maintained that trend for at least 3 years (that is, approximately halfway through the study) (Figure 1).

### 3.3. Changes in Energy, Nutrients, and Food Intake

Table 2 summarizes mean energy, nutrients, and food intake, according to the data collected in the FFQs, at baseline and after the first year of follow-up. In both studied groups (those who increased or decreased plant protein intake), modification of baseline dietary habits led to a significant reduction in the number of calories consumed per day, as well as cholesterol (all *p* < 0.001). Furthermore, in both groups, changes were accompanied by an increase in the intake of fruits and vegetables, with the subsequent increase in fibre consumption (all *p* < 0.01).

Patients who decreased plant protein intake augmented the amount of energy acquired from fats (MUFA and PUFA in particular), while reducing the amount of energy from carbohydrates (all *p* < 0.001). When evaluating dietary fatty acid profile (% total fat), a reduction of SFA and an increase of PUFA, with no changes in the proportion of MUFA, were observed. On the other hand, patients who increased plant protein consumption augmented carbohydrates as the source of energy at the expense of fats, reducing the intake of SFA and MUFA in particular (all *p* < 0.001). Regarding fatty acid profile, proportion of MUFA was maintained constant, whilst the proportion of SFA was reduced, and the proportion of PUFA was increased. Moreover, a significant increase in the intake of legumes (an important source of plant proteins) was observed in this group of patients (*p* < 0.001).

Since receiving nutritional guidance, the group of participants who increased plant protein intake consumed significantly less energy from fats and more from carbohydrates than the group which decreased plant protein intake (all *p* < 0.001). Consequently, the percentage of energy obtained from SFA, MUFA, and to a lesser extent PUFA was also reduced. Regarding the proportion of fatty acids in relation to the total amount of fat consumed, there were no differences between-groups in the proportion of MUFA, which was kept at about 54% of total fats. However, patients who increased plant protein intake exhibited a lower proportion of SFA and a higher proportion of PUFA compared to those who decreased plant protein intake (all *p* < 0.05). Finally, patients who increased plant protein intake consumed less cholesterol and more fibre and legumes (all *p* < 0.001). It is noteworthy to mention that, in both groups, the percentage of energy obtained from proteins was kept constant at around 18%, despite dietary intervention. The difference was that the increase in percentage of energy from plant proteins was done at the expense of reducing the percentage of energy from animal proteins and vice versa. A more detailed description of average intake of food groups that made up diets of participants according to the data collected in the FFQs at baseline and after the first year of follow-up is depicted in Appendix A.

### 3.4. Correlations between Changes in Energy, Nutrients, and Food Intake

To assess the interrelation between changes in dietary variables adjusted by energy, a Spearman correlation matrix was used (Figure 2). Changes in plant protein intake were positively correlated with changes in the intake of carbohydrates, fibre, and legumes and strongly negatively correlated with changes in the intake of SFA. Changes in plant protein intake were also mildly negatively correlated with changes in the intake of fats, MUFA, and proteins of animal origin.

### 3.5. Change in Plant Protein Intake and T2DM Incidence

The probability of T2DM incidence depending on the Δ of plant protein intake during dietary intervention was estimated using a Kaplan–Meier survival curve (Figure 3). To that end, patients were classified according to the median of Δ plant protein intake (median = 0.248%). We observed that patients who increased plant protein intake had lower probability of T2DM incidence than those who decreased plant protein intake, with a non-adjusted hazard ratio (HR) of 0.60 (0.41–0.89) (*p* = 0.0096).

Hazard ratios for T2DM incidence according to the median of Δ plant protein consumption are presented in Table 3. For the fully adjusted model, HRs (95% CI) was 0.64 (0.43–0.96) for those with Δ plant protein intake above the median, compared with those in the group below the median (*p* = 0.0024). Cox regression analysis also showed that T2DM incidence appeared to be associated with age (adjusted HR 1.03 (1.00–1.05)) and baseline BMI (adjusted HR 1.06 (1.01–1.11)).

## 4. Discussion

We hypothesized that a change in dietary habits towards a higher consumption of plant-based proteins, at the expense of proteins from animal sources, maintaining the percentage of energy received from proteins, could reduce the incidence of T2DM in a population with coronary heart disease. Using data from CORDIOPREV patients free of T2DM at baseline, we observed that a shift towards the consumption of more plant-based proteins was associated with 36% lower risk of developing T2DM. T2DM incidence in Spain has recently been estimated in 11.6 cases/1000 person-years (CI 95% = 11.1–12.1) [19]. However, we must underscore that our study population consists of coronary heart disease patients, which are at higher risk of suffering from T2DM than general population. In fact, incidence of T2DM in CORDIOPREV participants after a median follow-up of 5 years has been estimated in 58.1 cases/1000 person-years (CI 95%= 47.1–69.2), supporting the idea of how sensitive this population is and the urgent need for strategies to prevent or delay T2DM onset.

Regarding statistically significant differences found in Apo A1 and HbA1c levels, we reckon that these differences did not entail clinical relevance, because of the magnitude of the difference and because measurements were within reference values (Apo A1: 105–220 mg/dL, HbA1c: <6.5%). Nevertheless, it has been reported that the use of statins might affect HbA1c levels depending on the potency of statin and the duration of the treatment [20].

Due to insulinotropic effects of dietary proteins, high-protein diets have been proposed as a strategy to prevent T2DM onset. However, studies of the long-term effects of high-protein diets report conflicting results, as increasing dietary proteins, especially from animal origin, has been positively associated with an elevated risk of developing T2DM [21]. On the other hand, nutritional guidelines traditionally have set dietary reference values for protein based on nitrogen balance and essential amino acids content, identifying good quality proteins as the ones that supply sufficient indispensable amino acids [22]. However, as for the case of fats and carbohydrates, when studying long-term health outcomes of macronutrients consumption, it might be also necessary to focus on the source of proteins, because, in accordance with the previous classification, animal-based proteins would be generally graded as high-quality proteins, whereas plant-based proteins would be perceived as less nutritious and incomplete [23].

Studies that have investigated the importance of the source of protein in relation to T2DM incidence are scarce, and when it comes to plant protein intake, conclusions seem contradictory. In this regard, results from the EPIC-IntercAct case-cohort study described an association between elevated risk of T2DM and total and animal protein intake but not for plant protein intake [24]. Similarly, a cross-sectional study conducted in Harbin (China) found that higher intakes of total, animal, and red meat, but not plant protein, were associated with higher prevalence of T2DM in woman but not in men [25]. Researchers from the Melbourne Collaborative Cohort Study also reported that higher consumption of total and animal protein was associated with increased risk of T2DM, and plant protein intake was inversely associated with incident T2DM but only in women [26]. Finally, results from the nurses’ health study (I and II) and health professionals follow-up study highlighted, after comparing extreme quintiles, that percentage of energy intake from total and animal protein was associated with a higher risk of T2DM, and percentage of energy from plant protein was associated to a moderately decreased risk of T2DM in both sexes. The study also described a greater benefit on diabetes risk when replacing animal protein for plant protein [8]. On the other hand, Satija and collaborators (2019) created a healthful plant-based diet index that conferred positive scores to the intake of whole grains, vegetables, fruits, nuts, pulses, and vegetable oils, and negatively evaluated the intake of less healthy plant-based foods, such as fruit juices, refined grains, or animal foods [27], reporting an inverse relationship between the intake of healthy plant foods and risk of developing T2DM in three US cohorts. Results of the study support the idea of the need of comprehensive nutritional advice to not just recommend plant-based diets but to emphasize consumption of diets rich in healthy plant-based foods.

In the present study, we reported a significant reduction in T2DM incidence in the group of patients who increased the percentage of energy obtained from plant proteins, whose energy intake from plant protein was on average around 6%. These findings are in accordance with the results found in a meta-analysis carried out by Zhao and collaborators (2019) who observed the largest risk reduction of T2DM when the intake of energy from plant protein was about 6% [6].

Reducing SFA intake is a recommendation to decrease the risk of CVD. However, the role of fat quantity and quality in T2DM still remains unclear. In a recent systematic review and meta-analysis, Neuenschwander et al. (2020) found no or weak associations between total fat intake and the incidence of T2DM [28]. Similar results were presented by Liu et al. (2019) in a study carried out within the framework of the EPIC-NL study [29]. In a different meta-analysis, Imamura et al. (2016) found that replacing carbohydrates or saturated fats with unsaturated fats would help improve blood glucose control [30]. An inverse association between linoleic acid (which belongs to n-6 series of PUFA) intake and T2DM was reported by NHS, NHSII, and HPFS researches, particularly when linoleic acid replaced isocalorically SFA or carbohydrates [31]. Regarding carbohydrates, a study carried out in Sweden found inverse associations between intake of monosaccharides and fruits with T2DM, and positive associations for disaccharides and sweets [32]. Other studies have focused on glycemic index and glycemic load as the causal factors responsible for T2DM incidence [33,34], although it has also been suggested that glycemic index and glycemic load might be less useful tools to evaluate quality of carbohydrates than dietary fibre and whole-grain content [35]. Finally, in a study conducted by de Koning et al. (2011), it was found that low-carbohydrate diets were positively associated with T2DM risk when accompanied by high animal protein and fat [36], thus supporting the idea that, when it comes to evaluate the effects of limiting the intake of certain macronutrients, complexity of food matrices should also be considered.

In contrast to aforementioned observational studies, the CORDIOPREV study is an intervention trial, where patients have received comprehensive nutritional advice focused on the overall quality on the diet. Food choices are highly influenced by behavioural or social factors, and amid an obesogenic environment, penchant for high-energy density foods poses a challenge to fulfil an adequate adherence to a healthy dietary pattern. Here, we must underscore that the observed shift towards a healthier dietary pattern did not depend solely on participants free will, but on nutritional counselling provided by CORDIOPREV dietitians who, in the context of a controlled dietary intervention trial, encouraged the intake of legumes, whole grains, nuts (in Med diet), and vegetables, while discouraging red meat and processed meat, and that the overall long-term improvement and maintenance of adherence to two healthy diets (low-fat and Med diet) are possible to be accomplished with a comprehensive dietary intervention [16].

Food sources also have implications for nutrients uptake. In this sense, when assessing health benefits ascribed to the change in the consumption of plant proteins, not only nutrients that are being replaced must be considered, but also compounds that accompany proteins in the food matrix. Proteins are consumed in the context of what has been called the “protein package. This “package” includes abundant dietary fibre and micronutrients, what makes it complex to identify health benefits attributed solely to the presence of the studied ingredient [37]. However, as several studies have reported, in the general population, plant or animal protein intake seems to be strongly associated with diet quality, even when considering different dietary patterns. In fact, it has been described that plant protein in Western countries is a robust marker of nutrient adequacy of the diet [23], and protein choices are clearly associated with nutrient intake adequacy and quality [38]. In this regard, Shang and collaborators (2017) reported a positive correlation between animal protein and the consumption of SFA, and an inverse correlation with plant protein and fibre. They also described an inverse correlation of plant protein with SFA and MUFA [39]. In a study carried out in Canada, researchers found that red and processed meat contributed substantially to the intake of vitamin B12, zinc, MUFA, cholesterol, and SFA. Conversely, plant-based meat alternatives provided PUFA, MUFA, magnesium, and dietary fibre [40]. Similar results were also published by Phillips and collaborators (2015) [41]. Our results also support the idea of the synergistic effect of dietary components, as we found that an increase in the consumption of animal protein was positively correlated with SFA and cholesterol, and plant protein was positively correlated with fibre, legumes intake, and carbohydrates, and strongly negatively correlated with SFA.

Effects of dietary composition on metabolic pathways that regulate glucose and insulin secretion have been extensively reviewed elsewhere [42,43,44]. In this regard, dietary fibre interferes with carbohydrate and protein absorption and reduces postprandial glucose response, and food components such as polyphenols interact with molecular pathways related to glucose homeostasis. Similarly, replacing sources of animal protein with plant protein reduces organism supply of heme iron, advanced glycation end products, cholesterol, nitrate and nitrite, trimethylamine N-oxide, and branched chain amino acids, which have been shown to contribute to diabetes development. However, when it comes to the protective effects of plant proteins per se, the effect is not that clear, although some researchers have argued that the amelioration of T2DM risk is most likely down to their capability to improve body weight, blood pressure, blood lipids profile, and inflammatory markers [45]. On the other hand, it has been described that plant protein consumption can be associated with a particular profile of plasma metabolites which reflect modifications in metabolic pathways that might be involved in disease prevention or development [46].

The strengths of our study included that CORDIOPREV is a controlled dietary intervention trial tailored to evaluate direct impact of two different diets on the appearance of cardiovascular events. Other studies published on the topic are observational studies. The collection of diet data by trained dietitians in face-to-face interviews and the assessment of maintenance of dietary habits in time ensures the quality of the study. Furthermore, to minimize measurement errors associated with FFQs, diet outcome was evaluated using FFQs validated in a Spanish population who shared the same characteristics as our study population. Among the limitations of this study, we must acknowledge that it has been performed in a population of coronary heart disease patients from a Mediterranean area, meaning the generalizability of our findings to healthy people or people living in different areas may be limited. Secondly, this study uses data from a long-term, comprehensive dietary intervention; therefore, it might be possible that, without professional advice, free-living populations might not achieve the same level of compliance. Thirdly, T2DM incidence was not the primary endpoint of the CORDIOPREV, although it was a secondary objective of the trial. Finally, complex additive effect of different food components in diet makes it difficult to attribute the observed reduction in T2DM incidence to the sole change in plant protein intake with data obtained from FFQs. Thus, further molecular determinations would be needed to offer a complete explanation which join all the results showed in our study.

In conclusion, the present study shows that, in the context of the CORDIOPREV study, increasing the intake of plant-based proteins is associated with a reduction in T2DM incidence in a population of coronary heart disease patients at high risk of recurrence. We also show that enriching the diet with plant proteins instead of animal protein is accompanied by an improvement of the dietary pattern that cuts down the intake of SFA while increasing the amount of carbohydrates, fibre, and pulses. Herein, we provide evidence that healthy plant proteins must be encouraged at the expense of animal protein in nutrition therapy programs aimed to prevent T2DM onset in coronary heart disease patients.

## Figures and Tables

**Figure 1 nutrients-13-01217-f001:**
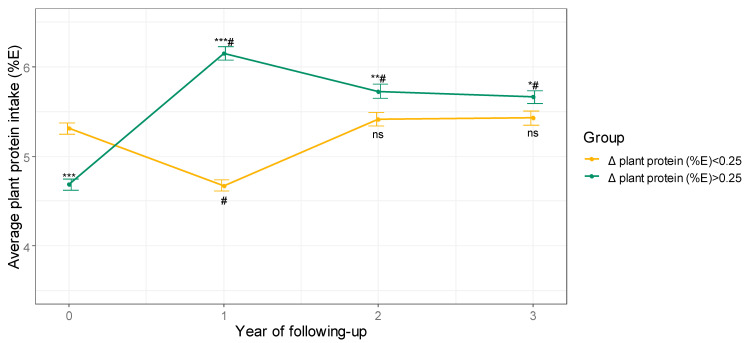
Average plant protein consumption during 3 years of follow-up. Values are expressed as mean ± SEM. *** *p* < 0.001, ** *p* < 0.01, and * *p* < 0.05 for comparisons between groups at each visit. # *p* < 0.001 for comparisons with baseline in each group. Δ, change; E: energy, ns: not significant.

**Figure 2 nutrients-13-01217-f002:**
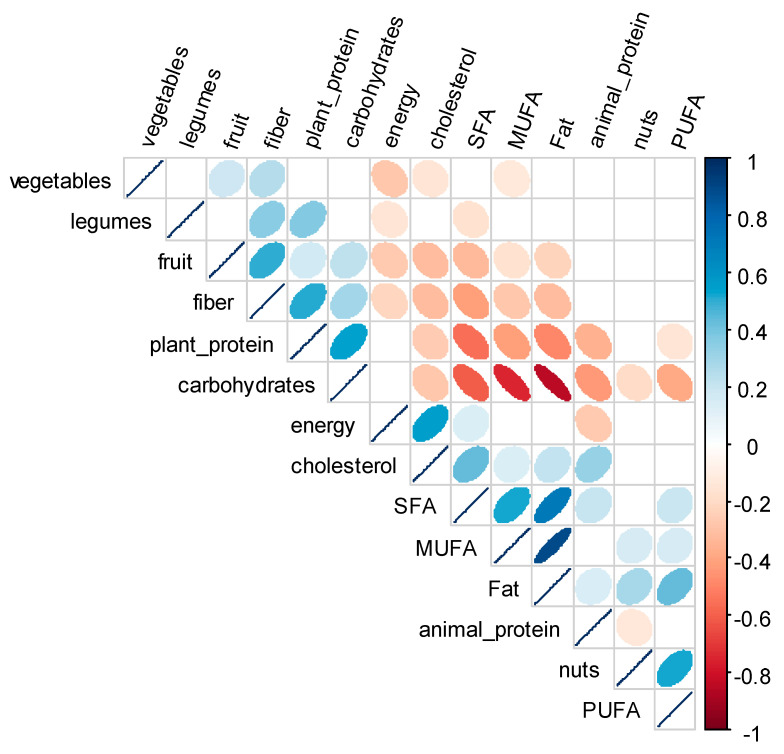
Correlogram of the upper triangular of the correlation matrix of the change in energy, nutrients, and food consumption (adjusted by energy) of 436 individuals from the CORDIOPREV (coronary diet intervention with olive oil and cardiovascular prevention) study after receiving dietary counselling. Figure shows correlations which *p*-values were <0.01. Positive correlations are displayed in blue and negative correlations in red, and colour intensity is proportional to the correlation coefficient. The ellipses have their eccentricity parametrically scaled to the correlation value. SFA: saturated fatty acids; MUFA: monounsaturated fatty acids; and PUFA: polyunsaturated fatty acids.

**Figure 3 nutrients-13-01217-f003:**
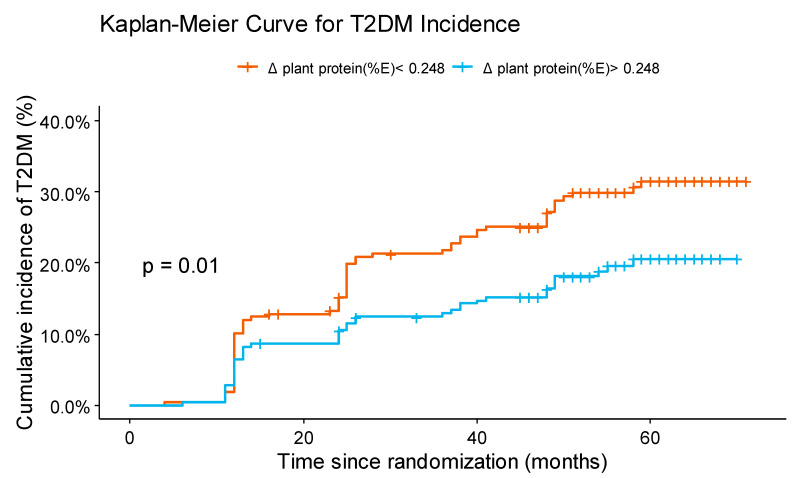
Cumulative incidence of type 2 diabetes mellitus (T2DM) (%). Cumulative incidence of T2DM (%) for two groups of patients: those with Δ in plant protein (%E) intake (changes produced between post- and pre-intervention) below the median and those above the median. Δ, change; T2DM: type 2 diabetes mellitus; E: energy.

**Table 1 nutrients-13-01217-t001:** Baseline characteristics of study participants according to the median of Δ in plant protein (percentage of energy (% E)) consumption.

Variable	Decreased Plant Protein Intake (<P_50_ [−3.22,0.248])	Increased Plant Protein Intake (>P_50_ (0.248,5.57])	*p* Value
*n*	218	218	
Age (years)	57.4 (0.6)	58.1 (0.6)	0.4315
Men/women (*n*)	182/36	187/31	0.5953
Waist circumference (cm)	103.1 (0.7)	101.8 (0.7)	0.1961
BMI (kg/m^2^)	30.6 (0.3)	30.1 (0.3)	0.2491
TG (mg/dL)	121.6 (4.2)	122.8 (4)	0.6761
Total cholesterol (mg/dL)	164 (2.4)	159.1 (2)	0.1673
c-LDL (mg/dL)	93.28 (1.81)	90.79 (1.61)	0.3964
c-HDL (mg/dL)	45.14 (0.77)	43.56 (0.66)	0.1420
Apo A1 (mg/dL)	136.4 (1.6)	131.5 (1.4)	0.0303
Apo B (mg/dL)	74 (1.3)	71.1 (1.1)	0.1306
hs-CRP (mg/L)	2.4 (0.1)	2.1 (0.1)	0.4369
Glucose (mg/dL)	94.1 (0.7)	92.5 (0.7)	0.1002
HbA1c (%)	5.94 (0.02)	5.85 (0.02)	0.0122
Insulin (mU/L)	9.05 (0.42)	8.5 (0.42)	0.3340
HOMA-IR	2.9 (0.1)	2.6 (0.1)	0.0148
Treatment with statins, %	83.7	90.4	0.0064
Hypertension, %	66.5	63.8	0.6153
Current smoking, %	8.25	6.9	0.7173
Prior smoking, %	67.0	70.2	0.5359
Diet (Low-fat/Med diet)	101/117	98/120	0.8475

Values expressed as mean (SEM). Δ, change; BMI, body mass index; TG: triglycerides; c-HDL: high-density lipoprotein cholesterol; c-LDL: low-density lipoprotein cholesterol; Apo A1: apolipoprotein A1; Apo B: apolipoprotein B; hs-CRP: high-sensitivity C-reactive protein; HbA1c: glycosylated haemoglobin; HOMA-IR: homeostasis model assessment-insulin resistance. Continuous variables were analysed using *t*-test or Wilcoxon rank sum test for unpaired data when data did not fit the normal distribution. Categorical variables were analysed using χ2 test.

**Table 2 nutrients-13-01217-t002:** Mean baseline and after 1 year of intervention values in energy, nutrient, and food intake.

Variable	Baseline	1 Year of Follow-Up		
	*n*= 436	Decreased Plant Protein Intake (<P_50_ [−3.22,0.248]) *n* = 218	Increased Plant Protein Intake (>P_50_ (0.248,5.57]) *n* = 218	Between-Group Differences Postintervention (*p*-Value)
Energy, kcal/d	2292.3 (24.1)	1942.4 (34.4) ***	1890.3 (25.3) ***	0.3523
Fat (%E)	36.1 (0.2)	36.9 (0.4) ***	32.4 (0.5) ***	<0.001
SFA (%E)	8.7 (0.1)	8.3 (0.1)	6.9 (0.1) ***	<0.001
SFA (% total fat)	26.8 (0.2)	24.92 (0.29) ***	23.94 (0.26) ***	0.0125
MUFA (%E)	17.7 (0.2)	18.3 (0.3) ***	15.8 (0.3) ***	<0.001
MUFA (% total fat)	54.2 (0.2)	54.15 (0.49)	53.65 (0.45)	0.4511
PUFA (%E)	6.2 (0.1)	6.9 (0.2) ***	6.5 (0.1)	0.0348
PUFA (% total fat)	19.0 (0.2)	20.94 (0.47) **	22.42 (0.42) ***	0.0186
Protein (%E)	18.2 (0.1)	18.3 (0.2)	18.0 (0.2)	0.3046
Vegetal protein (%E)	5.0 (0.0)	4.67 (0.06) ***	6.15 (0.08) ***	<0.001
Animal protein (%E)	12.40 (0.1)	12.87 (0.22) ***	11.31 (0.17) ***	<0.001
Carbohydrates (%E)	42.5 (0.3)	41.2 (0.5) ***	46.5 (0.5) ***	<0.001
Cholesterol (mg/d)	327.1 (4.5)	273.4 (6.1) ***	234.4 (4.5) ***	<0.001
Fibre, g/100 kcal	1.1 (0.0)	1.22 (0.02) **	1.48 (0.03) ***	<0.001
Fruit, g/100 kcal	16.3 (0.5)	21.63 (0.77) ***	22.36 (0.61) ***	0.1664
Vegetables, g/100 kcal	11.5 (0.2)	13.33 (0.44) **	13.51 (0.38) ***	0.5681
Legumes, g/100 kcal	1.0 (0.0)	1.1 (0.04)	1.46 (0.06) ***	<0.001
Tree nuts, g/100 kcal	0.4 (0)	0.41 (0.04)	0.46 (0.04)	0.2962

Values are expressed as mean (SEM). Δ, change; SFA: saturated fatty acids; MUFA: monounsaturated fatty acids; and PUFA: polyunsaturated fatty acids. Between-group differences were assessed using t-test or Wilcoxon rank sum test for unpaired data when data did not fit normal distribution, and within-group differences were assessed using paired t-test or Wilcoxon signed rank test when data did not fit normal distribution. Within-group differences from baseline: ** *p* < 0.01; *** *p* < 0.001.

**Table 3 nutrients-13-01217-t003:** Hazard ratios (95% confidence intervals) of T2DM incidence according to the median of Δ plant protein (%E) consumption.

	Increased Plant Protein Intake (>P_50_ (0.248,5.57])	Likelihood Ratio Test
Unadjusted model	0.6008 (0.4064–0.8883)	*p* = 0.0096
Multivariable model 1	0.5981 (0.4043–0.8848)	*p* = 0.0199
Multivariable model 2	0.6385 (0.4257–0.9578)	*p* = 0.0024

Cox regression models were used to assess the risk of T2DM according to the median of Δ plant protein (%E) consumption. Multivariable model 1 was adjusted for age, sex, and intervention group. Model 2 was further adjusted for prevalence of hypertension, baseline levels of HDL, triglycerides, and BMI, statin use, smoking status (never, former or current smoker), and alcohol intake. Δ: change; T2DM: type 2 diabetes mellitus; E: energy.

## Data Availability

Data are available upon request for researchers who meet the criteria for access to confidential data. Inquiries are to be addressed to: Prof. José López-Miranda (corresponding author) (contact via: jlopezmir@uco.es) and/or Committee ethics of Reina Sofia University Hospital of Cordoba (contact via: cetico.hrs.sspa@juntadeandalucia.es).

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
