# Peer review of "Quality and Quantity of Protein Intake Influence Incidence of Type 2 Diabetes Mellitus in Coronary Heart Disease Patients: From the CORDIOPREV Study"

_nutrients, 2021, doi:10.3390/nu13041217_

Round 1
Reviewer 1 Report
The objective of this well written manuscript was to assess whether consumption of more plant-based proteins, at the expense of animal proteins are associated with a reduction of T2DM incidence in a population with coronary heart disease. Additional information on diets and further discussion of obtained results will improve the manuscript. Please see below for some comments:
Line 89-99: It would be useful to include a chart with more details on the diets and composition. For example, what was the sources of these macronutrients? What type of protein was used? (there are a lot of differences in the functions of different plant proteins). The same thing is true for the source of carbs and fats.
Table 1: Why Apo A1 and HbA1c are reduced in the group with higher plant protein and what it means? These all need to be discussed in the discussion.
Table 2: The majority of results given in this Table have not been discussed. For examples having a lower PUFA and MUFA for the group with increased protein intake is suggestive of what? What others have found on these? The same thing is true for the other markers listed here and change significantly between these two groups.
Author Response
Thank you for reviewing our manuscript and for constructive criticism on the paper. The comments and suggestions you have made have helped us to improve the quality of the work. Please find below and itemized list of changes and detailed answers to every point raised in your reviews.
As requested by the editor, ethic approval number of CORDIOPREV study has been included in the manuscript (Line 66).

Reviewer 2 Report
Authors evaluated the association between increase/decrease in plant protein intake and incidence of type 2 diabetes mellitus in patients with coronary heart disease, included in the trial “CORDIOPREV study”. Results demonstrated that patients that increased plant protein intake had a lower risk of developing type 2 diabetes, after a median follow-up of 60 months (HR=0.64, (0.43-0.96)).
Paper is well written and results are interesting. Manuscript would benefit from few clarification and amendments.
1 - At page 1 line 39 consider to write “increased physical activity” and not “increase physical activity”.
2 - Please add reference for the sentences of paragraph between line 48 and 50 at page 2.
3 – At paragraph 3.2 please rephrase and clarify the first three lines. Patients who decreased plant protein intake, modified the amount of proteins from baseline or from value at the end of first years of follow up?
4 – Patients who decreased plant protein intake augmented the amount of energy from fat while patients who increased plant protein intake augmented amount of carbohydrates, fibre, and legumes. Reduced incidence of type 2 diabetes could be due to this complex differences in nutritional components of diet than to protein intake in itself. Is it possible to include variables of nutritional components of diet (intake of carbohydrates, fibre, fat) in the regression models for risk of type 2 diabetes? Otherwise discuss in detail about this concept.
5 – Physical activity has a key role in amelioration of lifestyle habits and in reducing risk of metabolic diseases. Can the authors add some information about physical activity in the study population?
6 – It would be useful to report incidence of type 2 diabetes in general population and in population at risk for cardiovascular disease of the country in which the study took place. This could help to make comparison between study results and epidemiological data.
Author Response

(The authors gave the same response as above.)

Round 2
Reviewer 1 Report
Authors have addressed the majority of my concerns.